

# Microbial community diversity patterns are related to physical and chemical differences among temperate lakes near Beaver Island, MI

Miranda H. Hengy, Dean J. Horton, Donald G. Uzarski and Deric R. Learman

Institute for Great Lakes Research and Department of Biology, Central Michigan University, Mount Pleasant, MI, United States of America

## ABSTRACT

Lakes are dynamic and complex ecosystems that can be influenced by physical, chemical, and biological processes. Additionally, individual lakes are often chemically and physically distinct, even within the same geographic region. Here we show that differences in physicochemical conditions among freshwater lakes located on (and around) the same island, as well as within the water column of each lake, are significantly related to aquatic microbial community diversity. Water samples were collected over time from the surface and bottom-water within four freshwater lakes located around Beaver Island, MI within the Laurentian Great Lakes region. Three of the sampled lakes experienced seasonal lake mixing events, impacting either $O_2$, pH, temperature, or a combination of the three. Microbial community alpha and beta diversity were assessed and individual microbial taxa were identified via high-throughput sequencing of the 16S rRNA gene. Results demonstrated that physical and chemical variability (temperature, dissolved oxygen, and pH) were significantly related to divergence in the beta diversity of surface and bottom-water microbial communities. Despite its correlation to microbial community structure in unconstrained analyses, constrained analyses demonstrated that dissolved organic carbon (DOC) concentration was not strongly related to microbial community structure among or within lakes. Additionally, several taxa were correlated (either positively or negatively) to environmental variables, which could be related to aerobic and anaerobic metabolisms. This study highlights the measurable relationships between environmental conditions and microbial communities within freshwater temperate lakes around the same island.

## INTRODUCTION

Lakes are complex ecosystems that span a range of physical and chemical properties, which are driven by differences in formation, hydrology, weather patterns, and geology (*Wetzel, 2001*). Further, even lakes within the same geographic region can vary widely in physicochemical conditions, both spatially and temporally based on formation, age, and trophic status (*Clement, Murry & Uzarski, 2015*). The physical and chemical attributes of a lake can impact microbial communities and the biogeochemical processes they

Corresponding author
Deric R. Learman,
deric.learman@cmich.edu

mediate, since microbial communities are governed by local environmental conditions. The essential processes regulated by microbial communities include, but are not limited to, nutrient cycling (e.g., carbon, nitrogen, and sulfur), which supports biologically suitable environmental conditions within lakes (*Essington & Carpenter, 2000*), as well as chemical export, such as respiration of $CO_2$, and other redox-sensitive elements (*Paerl & Pinckney, 1996*; *Pilcher et al., 2015*). As microbial community function is related to microbial community composition (*Bier et al., 2015*), and community composition is constrained by local environmental conditions, it is important to explore microbial communities within individual lakes.

While environmental conditions are unique to each lake, environmental gradients can also occur within some lakes that physicochemically stratify. Water column mixing, or turnover, followed by a return to stratified conditions is a natural ecosystem disturbance that occurs seasonally in many lakes. This phenomenon is known to influence microbial communities, as a consequence of shifting environmental conditions, and even impacts microbial community assembly mechanisms (*Tammert, Kisand & Nõges, 2005*; *Shade, Jones & McMahon, 2008*; *Shade, Chiu & McMahon, 2010a*; *Shade, Chiu & McMahon, 2010b*; *Shade et al., 2011*; *Shade et al., 2012b*; *Garcia et al., 2013*; *Meuser et al., 2013*; *Andrei et al., 2015*). The stratification of water masses at different temperatures and densities results in a hypolimnion that is not only colder, but tends to have lower dissolved oxygen and pH relative to the epilimnion as the rate of decomposition tends to exceed photosynthesis (*Fenchel & Finlay, 2008*). Furthermore, inorganic nutrients (e.g., C, N, and P) may accumulate in the hypolimnion (*Tõnno, Ott & Nõges, 2005*; *Zadereev, Tolomeev & Drobotov, 2014*). Lake mixing events can transport dissolved organic carbon (DOC; described as the amount of C within a system) throughout lakes (*Mostofa et al., 2005*; *Kim, Nishimura & Nagata, 2006*; *Li et al., 2008*), and dissolved organic matter (DOM; quality of organic matter as described in *Chappaz & Curtis, 2013*) has previously been shown to vary between upper and lower layers of lakes (*Mostofa et al., 2005*). This suggests that structurally different organic compounds may not only differ among lakes, but also characterize each layer in some lakes. In addition, both DOC and DOM have been found to shape microbial community composition depending upon carbon source and concentration (*Cotner & Biddanda, 2002*; *Burkert et al., 2003*; *Crump et al., 2003*; *Eiler et al., 2003*; *Grossart et al., 2008*; *Amaral, Graeber & Calliari, 2016*; *Lucas et al., 2016*). As previously stated, chemical and physical components are major drivers of bacterial community structure and population shifts, therefore, lake stratification can present a major disturbance for bacterial communities and may impact microbial communities structure as lakes gradually stratify post-mixing.

Research to date demonstrates that microbial communities respond to disturbance with various degrees of resistance and resilience, depending upon the existing community and qualities of the disturbance (*Allison & Martiny, 2008*; *Shade et al., 2011*). For example, microbial communities may show resistance to lake mixing and physicochemical stratification, remaining unaffected in the face of disturbance (*Shade et al., 2012a*). However, depending upon the physicochemical attributes disturbed (e.g., $O_2$, nutrients, pH, specific conductance etc.), disturbance influences microbial communities differentially
in extent of community change, resistance, and resilience (*Shade et al., 2011*). Additionally, different subsets of microbes within a community (e.g., generalist vs. rare taxa) can experience different patterns of resistance and resilience. Illustrating this, *Shade, Chiu & McMahon (2010b)* found that many generalist taxa are resistant to mixing and subsequent changes of temperature and dissolved oxygen levels. Nevertheless, individual taxa (often specialist or rare) can be positively or negatively influenced as a result of physicochemical shifts and show fundamentally different reactions to mixing than dominant community members (*Shade, Chiu & McMahon, 2010a*; *Shade, Chiu & McMahon, 2010b*). As such, microbial communities can vary between lakes due to differences in lake chemistry, as well as within lakes at finer scales for the same reason.

In this study, three freshwater inland lakes of Beaver Island, Michigan, USA, as well as an adjacent location within Lake Michigan, were sampled to evaluate the relationship between microbial communities and local physicochemistry within surface-water and bottom-water habitats (epilimnion and hypolimnion during stratification). These lakes were selected as they each hosted unique and contrasting physicochemical properties (*Clement, Murry & Uzarski 2015*). Two of the lakes were holomictic and experienced oxygen stratification, while another holomictic lake (Lake Michigan) did not experience stratification at the point of sampling, but did experience a thermocline. The final lake (Barney's Lake) is a shallow lake which did not experience a mixing event and lacked physicochemically stratified layers. Specifically, we sought to explore the relationships between microbial community diversity and environmental variables known to stratify within lakes. We also explored microbial community diversity change over time within each lake with respect to post-mixing stratification of environmental variables or a lack thereof. Physical and chemical parameters were collected in conjunction with high resolution microbial community data (via 16S rRNA gene sequencing) to explore relationships between microbial taxa and natural physicochemical gradients among and within sampled lake systems.

## METHODS

### Sampling locations

Three inland lakes on Beaver Island, MI (Fox Lake [FL], Barney's Lake [BL], and Lake Geneserath [LG], located on Beaver Island, MI) and Lake Michigan (St. James Harbor [LM]; Fig. 1) were sampled during three collection periods in the summer of 2014: period 1 (June 10–11), period 2 (July 28–30), and period 3 (Aug. 30–31). Sampling sites for the three inland lakes were located at the region of greatest depth (at 3.6 m for Barney's Lake, 15.2 m for Lake Geneserath, and 6.1 m for Fox Lake). Lake Michigan bottom sampling depth ranged from 14.5–18.3 m, depending upon small-scale spatial bathymetric differences. While Lake Michigan was not sampled at the point of greatest depth (as were other lakes in this study), we attempted to sample Lake Michigan to a similar depth as inland lakes within this study.

Surface and bottom-water samples were collected using a Kemmerer (Wildco®, Yulee, FL, USA) water sampler. During each collection period, samples were retrieved from surface (one meter below the surface) and bottom-water (one meter above the lake sediment)

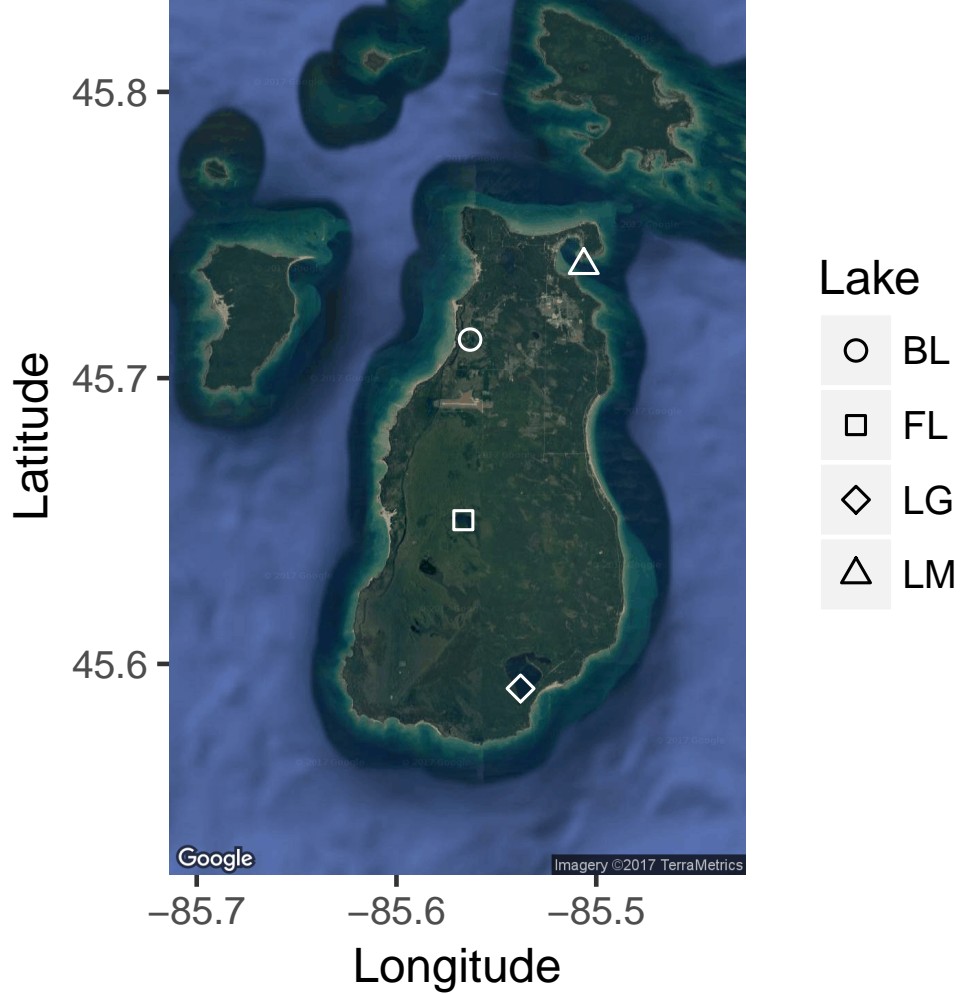

**Figure 1 Map of sampling region.** Map of sampled lakes in the Beaver Island region. Shapes correspond to lake sampled. (Map data© 2017 Google; 2017 TerraMetrics, Inc., http://www.terrametrics.com).

locations for each site. For each sample, water was collected in an acid washed sterile bottle. From this bottle, 90 ml of water was filtered through a 0.45 μm filter (Whatman, GE Healthcare, Little Chalfont, Buckinghamshire, UK) into acidified vials (resulting pH of 3) and stored on ice for DOC analysis. For collection of microbial samples, 120 ml of water was syringe filtered through a combination of two filters (2.2 μm first, followed by 0.22 μm filters). The filters were flash frozen (dry-ice and ethanol bath) in the field, and then stored at −80 °C. Once per sampling period, 120 ml of sterilized Nanopure water was filtered and frozen in the field as a control for microbial samples. The remaining water was stored on ice and then filtered (0.45 μm) in the lab for nutrient analyses.

A calibrated Hydrolab® DS5 (OTT Hydromet, Kempten, Bavaria, Germany) was used to generate a physicochemical profile of each lake prior to sample collection. Measured

parameters included dissolved oxygen (percent and mg/L), temperature (°C), and pH (raw data can be found in Table S1).

## Water chemistry analyses

For nutrient analysis, 250 ml of water from each sampling location and depth was filtered in the lab through a 0.45 μm filter (Whatman) and frozen at −20 °C. A Quaatro Bran+Luebbe Auto Analyzer with an XY-2 Sampler (Seal Analytical, Mequon, WI, USA) was used to determine soluble reactive phosphorus (SRP), ammonium ($NH_4$), nitrate ($NO_3^-$), total nitrogen (TN), and total phosphorus (TP) concentrations in the water. An additional 10 ml of water was filtered (0.45 μm) and acidified for dissolved organic matter (DOM) analysis. Proxies of DOM were characterized by their specific absorption coefficient (SAC340) (*Chappaz & Curtis, 2013*; *Curtis & Adams, 1995*) and specific UV absorbance (SUVA254) (*Mcknight et al., 2001*; *Weishaar et al., 2003*). Triplicates of each sample were placed into quartz cuvettes (1 cm width) and UV absorbance readings were taken at two different wavelengths: 254 nm and 340 nm. Samples collected for DOC analysis (described above) were quantified using a Shimadzu TOC-V analyzer (Kyoto, Japan). Raw water chemistry data can be found in Table S2.

## Microbial taxonomic analysis

DNA was extracted from frozen filters using the MoBio PowerWater® DNA isolation kit (following the manufacturer's protocol). DNA was extracted from both .22 and 2.2 μm filters from the same sample simultaneously. All samples were concentrated in a Zymo DNA Clean & Concentrator™ kit before being quantified by a Qubit® 2.0 Fluorometer (Life Technologies, Carlsbad, CA, USA). Control samples yielded DNA that was below detection limits (<0.5 ng/mL). In order to obtain a sufficient amount of DNA for downstream sequencing, PCRs were completed for each sample to amplify the 16S rRNA gene using high-fidelity *Taq* polymerase (New England BioLabs Inc., Ipswich, MA, USA) and 27F and 1492R primers (*Weisburg et al., 1991*). PCR conditions implemented were as follows: initial denaturation at 95 °C for 5 min, followed by 36–40 cycles (denaturation at 95 °C for 30 s, annealing at 56 °C for 30 s, and extension at 72 °C), and final extension at 72 °C for 10 min. The number of cycles for each sample varied due to differences in amplification (Table S3), which was visualized through gel electrophoresis. Replicate PCRs for each sample were pooled. PCR samples were purified using the QIAquick® Gel Extraction Kit (Qiagen, Hilden, North Rhine-Westphalia, Germany). Three sampling points were excluded from microbial community data analysis, which included bottom-water time point "1" for Lake Michigan and both surface and bottom-water community profiles for Fox Lake time point "3". These samples were excluded, as they either did not contain sufficient concentration or quality of DNA for sequencing or analysis. V4 16S rRNA amplicons were generated using previously described methods and primers 16Sf-V4 (515f) and 16Sr-V4 (806r) (*Kozich et al., 2013*) and sequenced on an Illumina MiSeq platform using a paired end 2 × 250 bp format (accomplished by Michigan State University's Research Technology Support Facility).

Sequence data were processed using MOTHUR v.1.35.1 (*Schloss et al., 2009*). Quality control and clustering steps were implemented following the publicly available MiSeq SOP

(found at http://www.mothur.org/) with modifications. Briefly, sequences which were less than 251 bp or greater than 254 bp in length were removed from further analyses, as were sequences which contained >8 homopolymers. Sequences were aligned using the SILVA (v. 119) reference database (*Quast et al., 2012*). Sequences which were not aligned within the V4 region were also removed. UCHIME (*Edgar et al., 2011*) was used to check for chimeric DNA, which was subsequently removed. Sequences were classified using the RDP database (training set v9; *Cole et al., 2013*). Classifications corresponding to chloroplast, eukaryotic, or mitochondrial DNA, as well as sequences that classified as unknown, were removed. The remaining data were clustered into operational taxonomic units (OTUs) using a 0.03 dissimilarity threshold. The Mothur workflow associated with this study can be found within an online repository located on GitHub (https://github.com/horto2dj/CMUBS_microb). Sequences obtained for this study have been deposited in the MG-RAST database (*Meyer et al., 2008*) under accession numbers mgm4732740.3–mgm4732751.3, mgm4732757.3, mgm4732760.3, mgm4733677.3–mgm4733686.3, mgm4733688.3, mgm4733690.3–mgm4733704.3, and mgm4733784.3–mgm4733785.3. Additional metadata associated with submitted environmental sequences can be found within Table S3.

## Statistical analyses

Statistical analyses (both chemical and biological) were completed using the R statistical software v.3.2.1 (*R Core Team, 2015*). Protocols and files associated with quality control and statistical tests can be found on GitHub (https://github.com/horto2dj/CMUBS_microb). Differences in lake chemistry among lakes and time points within lakes were analyzed through principal component analysis (PCA).

Prior to alpha and beta diversity analyses, singletons and doubletons were removed and samples were normalized using the *DeSeq2* package (*Love, Huber & Anders, 2014*) in R, followed by a variance stabilizing transformation (*McMurdie & Holmes, 2014*).

Using the *PhyloSeq package* (*McMurdie & Holmes, 2013*), Shannon's diversity was calculated for microbial communities of each sample. Linear mixed-effect models (with 'Lake' as random effect) and ANOVA were used to test significance of habitat (i.e., surface vs bottom-water) on levels of alpha diversity. Linear models and ANOVA were used to test for differences in alpha diversity between lakes. Alpha diversity values were correlated with measured environmental variables using Spearman's rank correlation to explore relationships between environmental variables and alpha diversity.

Non-metric multidimensional scaling (NMDS) based on Bray–Curtis distance was performed to compare dissimilarity between the samples, also employing the *PhyloSeq* package. A total of 20 iterations were accomplished to reach the lowest stress during NMDS and two dimensions ($k = 2$) were used for visualization. Analysis of Similarity (ANOSIM) was used to test for significant differences in community composition between microbial communities of different lakes. Correlation of environmental variables with microbial communities was determined using *envfit* of the Vegan package (*Oksanen et al., 2017*). Canonical Correspondence Analysis (CCA) was implemented to explore relationships between environmental variables significantly correlated to beta diversity in NMDS and microbial community beta diversity. Permutation tests were implemented to

test significance of axes and environmental variables within CCA in explaining microbial community beta diversity patterns using 999 permutations in all tests. Partial Canonical Correspondence Analysis (pCCA) was implemented to specifically examine potential effects of oxygen gradients on microbial communities in the same way as described above.

Spearman's Rank correlations were used to identify OTUs significantly correlated to environmental variables (i.e., dissolved oxygen, pH, and temperature). Only OTUs which appeared within a minimum of five samples with at least two sequences were considered for correlation analyses. Variance stabilizing transformation was used to normalize sequence abundances across samples for these OTUs to account for uneven sequencing depth between samples. Correlations with $p > 0.001$ and $r < 0.65$ were excluded as an attempt to reduce spurious correlations. OTUs which could not be identified as belonging to a phylum were removed from analyses.

## RESULTS & DISCUSSION

### Physicochemical variation among and within lakes

Fundamental differences in lake physicochemistry were observed between Lake Michigan and inland lakes on Beaver Island (Fig. 2; Table 1). Lake Michigan water chemistry was distinguished based upon DOC, $NO_3^-$, and SAC340 concentrations, showing considerable divergence from the remaining three lakes (Fig. 2). Lake surface physicochemistry (temperature, DOC concentrations, and DOM properties) was nearly indistinguishable between Barney's Lake and Lake Geneserath. Fox Lake surface water chemistry was also similar to that of Barney's Lake and Lake Geneserath, but was slightly dissimilar due to a lower pH with respect to other lakes.

Of the lakes sampled, Lake Geneserath and Fox Lake experienced oxygen, temperature, and pH stratification over time between surface and bottom-waters (Fig. 2, Table S2). The bottom-water in both lakes experienced lower temperatures, elevated acidity, and lower oxygen levels with respect to the surface-water. DOC concentrations and DOM quality did not vary significantly (two-tailed $t$-test, $p > 0.05$) between surface-water and bottom-water environments for any lake. Barney's Lake and Lake Michigan did not experience physicochemical stratification at the points sampled. However, Lake Michigan bottom water experienced decreased temperatures, but did not plateau with increasing depth, suggesting that the thermocline rather than the hypolimnion was developed at the sampling location.

### Microbial community taxonomy and alpha diversity among lakes

A total of 3,415,100 sequences were obtained across all samples prior to filtering and quality control steps. After quality filtering steps, 2,058,143 sequences remained and from these sequences 51,831 OTUs were identified. Sequencing depth ranged from 54,802 to 136,518 total sequences among samples. After singletons and doubletons were removed, a total of 20,372 OTUs remained for diversity analyses. There were no significant differences in alpha diversity among lakes according to linear models and ANOVA. However, linear mixed-effect models and ANOVA found that habitat type (i.e., surface vs bottom-water) significantly influenced Shannon diversity levels ($p < 0.01$), with higher levels of diversity
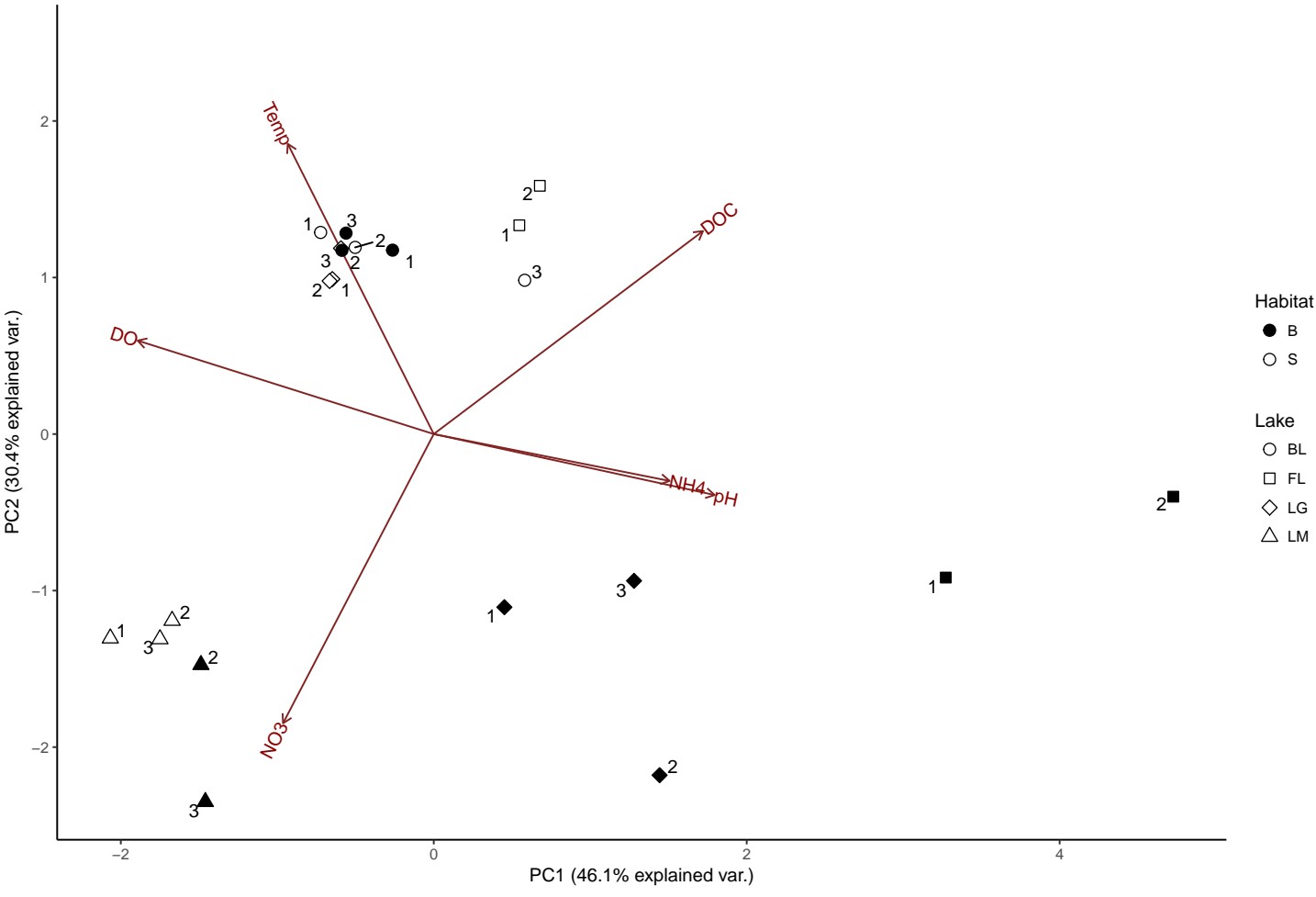

**Figure 2** **Principal Component Analysis (PCA) demonstrating separation of all sampling points based on measured environmental variables (pH, temperature, nitrate, ammonium, nitrate, dissolved oxygen, and dissolved organic carbon).** Circles, Barney's Lake; squares, Fox Lake; diamonds, Lake Geneserath, triangles, Lake Michigan. Open shapes correspond to surface-water samples, while filled shapes correspond to bottom-water samples. Numbers associated with points correspond to time point of sampling (higher numbers are later in the summer). DO, dissolved oxygen; DOC, dissolved organic carbon; NH4, ammonium; NO3, nitrate; Temp, temperature. SAC340 and $NO_3^-$ data were highly correlated (Spearman Rank, $r = 0.82$, $p < 0.001$) so both were represented by ''NO$_3$'' in the PCA.

occurring in bottom-water habitat versus the surface-water (Table 2). Previous literature that suggests anoxic hypolimnion communities are more diverse (alpha diversity) than their respective epilimnion (*Humayoun, Bano & Hollibaugh, 2003*; *Shade et al., 2012b*; *Meyerhof et al., 2016*), which is consistent with our findings in lakes which developed anoxic hypolimnia (Fox Lake and Lake Geneserath). Two of the lakes within this study, Barney's Lake and Lake Michigan, did not develop anoxic hypolimnia, yet these systems experienced higher alpha diversity in their bottom-water environments with respect to surface waters. These differences in alpha diversity (namely evenness) between surface and bottom-water environments may be driven by other variables, such as temperature (in Lake Michigan), or other variables not measured in this study, such as light penetration.
**Table 1 Limnological characteristic ranges for the surface and bottom water of each lake during the duration of this study.**

|  | Habitat | Temp (°C) | pH | DO (%) | DOC (mg/L) |
|---|---|---|---|---|---|
| Barney's Lake | Surface | 20.7–21.7 | 8.57–8.75 | 103.1–118.7 | 10.7–11.4 |
|  | Bottom | 20.4–21.2 | 8.52–8.75 | 103.7–113.9 | 10.6–11.2 |
| Fox Lake | Surface | 20.7–21.5 | 6.13–6.53 | 90.7–93.8 | 16.0–17.6 |
|  | Bottom | 11.1–12.5 | 5.46–5.79 | 0–25.4 | 16.2–18.5 |
| Lake Geneserath | Surface | 19.7–21.4 | 8.05–8.29 | 97.8–103.4 | 9.2–9.8 |
|  | Bottom | 8.8–9.1 | 6.49–6.78 | 0–64.0 | 9.0–9.2 |
| Lake Michigan | Surface | 16.1–18.2 | 8.12–8.28 | 102.1–124.5 | 2.4–2.8 |
|  | Bottom | 8.1–15.9 | 7.84–8.08 | 103.4–120 | 2.1–2.6 |

Notes.
Temp, temperature; DO, dissolved oxygen; DOC, dissolved organic carbon.

**Table 2 Shannon diversity values for microbial communities from each collection point.**

| Lake | Habitat | Time | Shannon |
|---|---|---|---|
| BL | B | 1 | 4.21 |
|  |  | 2 | 4.83 |
|  |  | 3 | 4.32 |
|  | S | 1 | 3.59 |
|  |  | 2 | 4.07 |
|  |  | 3 | 4.27 |
| FL | B | 1 | 4.19 |
|  |  | 2 | 4.91 |
|  | S | 1 | 3.22 |
|  |  | 2 | 4.72 |
| LG | B | 1 | 4.01 |
|  |  | 2 | 4.33 |
|  |  | 3 | 4.67 |
|  | S | 1 | 3.73 |
|  |  | 2 | 4.7 |
|  |  | 3 | 4.29 |
| LM | B | 2 | 4.49 |
|  |  | 3 | 4.91 |
|  | S | 1 | 3.48 |
|  |  | 2 | 3.74 |
|  |  | 3 | 3.18 |

A separate study exploring microbial communities along a Lake Michigan transect south of our sampling location did not find differences in alpha diversity between epilimnion and hypolimnion environments (*Fujimoto et al., 2016*). However, as we did not sample the hypolimnion of Lake Michigan in our study, our results are not directly comparable to the findings of *Fujimoto et al. (2016)*. Despite this, we found differences between Lake Michigan epilimnion and thermocline environments, which suggests potentially higher diversity within the thermocline with respect to the surface water environment.
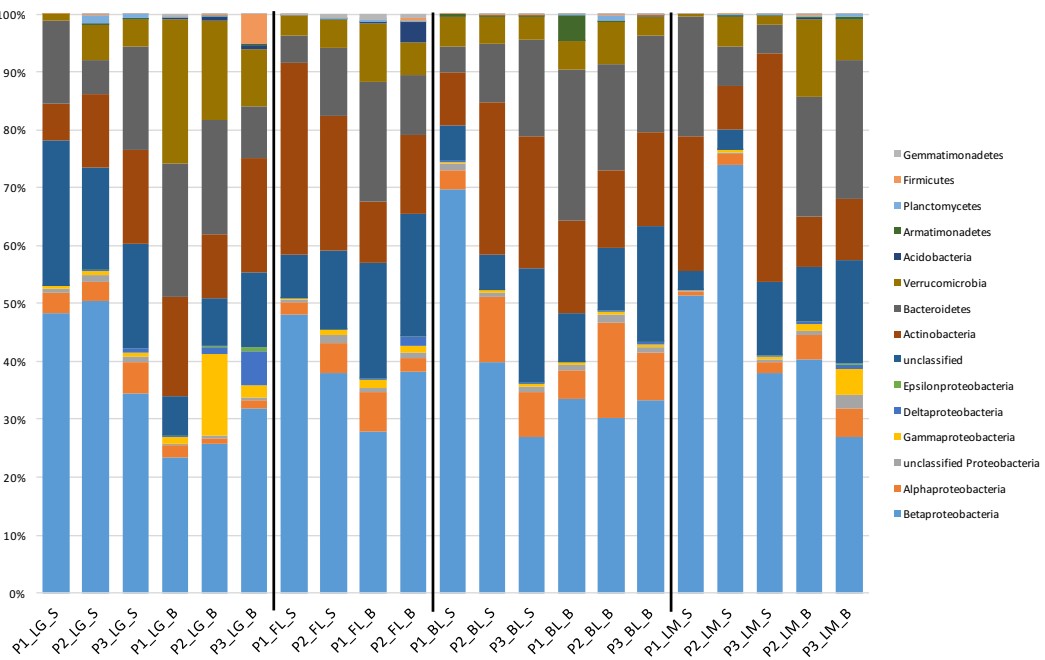

**Figure 3** **Taxonomic composition and relative abundance (>1% relative abundance) of community members broken down by phylum at each sampling location.** Labeling scheme is represented by lake (e.g., LG, FL, BL, and LM), followed by habitat (e.g., surface-water [S] and bottom-water [B]), and finally sampling time point (e.g., P1, sampling time 1, P2, sampling time 2, etc.).

Taxonomically, members of *Proteobacteria*, primarily *Betaproteobacteria*, were generally the most dominant taxa (based on relative abundance) found within the sequenced microbial community in all the explored lakes (Fig. 3). Other dominant phyla (>1% community composition) within the lake systems included *Acidobacteria, Actinobacteria, Armatimonadetes, Bacteroidetes, Firmicutes,* G*ammatimonadetes, Planctomycetes,* and *Verrucomicrobia.* These phyla have frequently been shown to dominate freshwater communities (*Attermeyer et al., 2015*; *Boucher, Jardillier & Debroas, 2006*; *Taipale, Jones & Tiirola, 2009*; *Zwart et al., 2002*). The most abundant OTU within the inland lakes, and second most abundant in Lake Michigan, was related to *Polynucleobacter* within *Betaproteobacteria.* This microbial genus has been commonly found in freshwater systems, with levels up to 60% community composition found in one freshwater pond (*Hahn, 2003*; *Hahn, Pockl & Wu, 2005*; *Hahn et al., 2010*; *Jezbera et al., 2011*) and represented the third most dominant OTU of another stratified lake (*Garcia et al., 2013*).

## Environmental relationships with microbial beta diversity

Beta diversity ordinations incorporating all sites showed microbial communities separated based on the sampling location (or lake) (Fig. 4; ANOSIM $R = 0.789, p = 0.001$). Significant relationships ($p < 0.001$) were found between environmental conditions and microbial beta diversity, including correlations between community structure and dissolved oxygen ($r = 0.645$), dissolved organic carbon ($r = 0.790$), pH ($r = 0.593$), and temperature

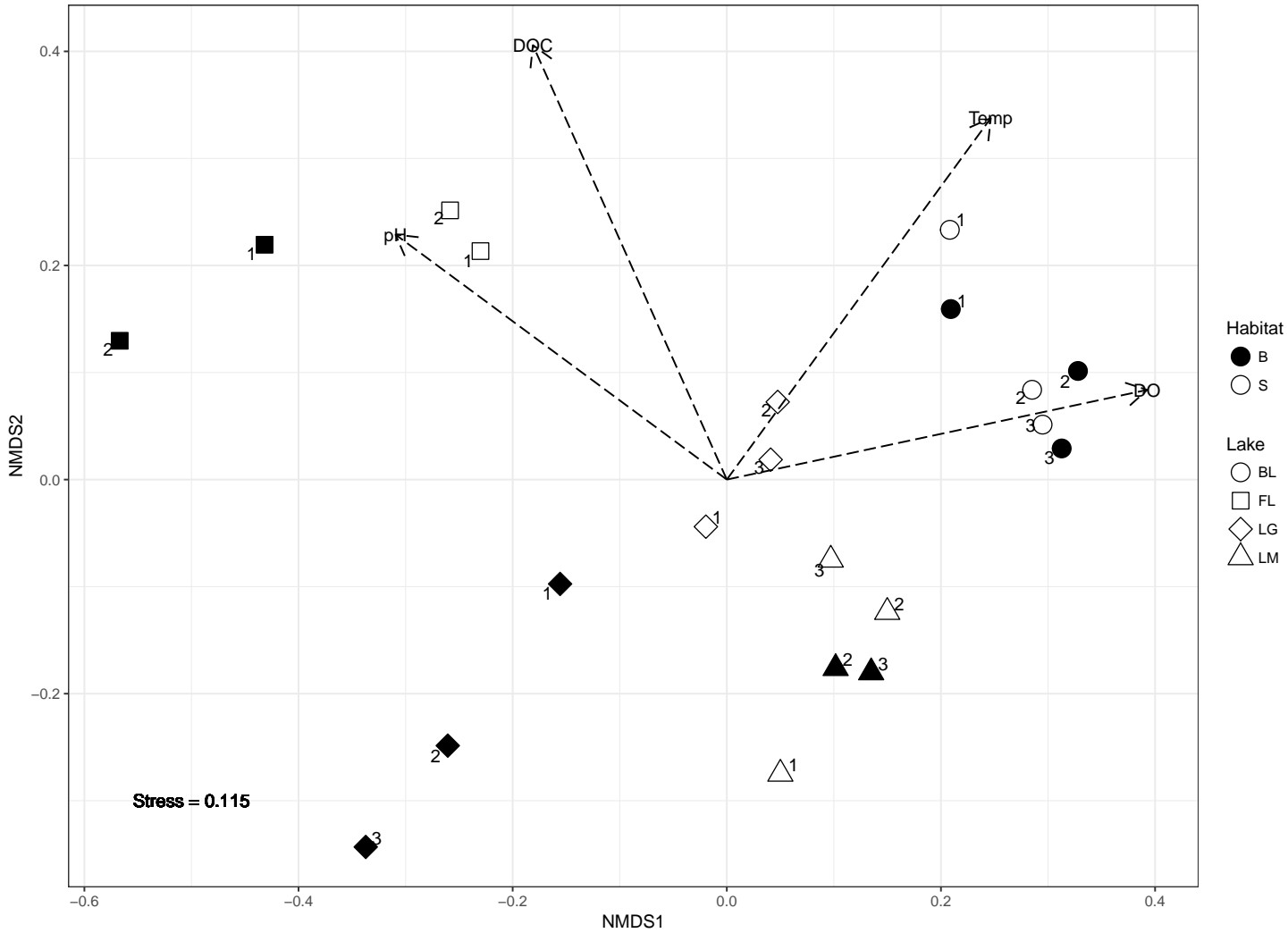

**Figure 4** **Non-metric Multidimensional Scaling (NMDS) of microbial communities.** NMDS of microbial communities within each sampled lake. Circles, Barney's Lake; squares, Fox Lake; diamonds, Lake Geneserath, triangles, Lake Michigan. Open shapes correspond to surface-water samples, while filled shapes correspond to bottom-water samples. Numbers associated with points correspond to time point of sampling. Vectors correspond to environmental variables significantly correlated ($p < 0.001$, $R > 0.5$) to separation of microbial communities within NMDS. DO, dissolved oxygen; DOC, dissolved organic carbon; Temp, temperature.

($r = 0.699$). Environmental variables found to significantly correlate to beta diversity in NMDS (i.e., DO, DOC, pH, and temperature) were tested as constraining variables on beta diversity. CCA was found to be significant ($F = 1.4245$, $p < 0.001$) in explaining microbial beta diversity among all samples (Fig. S1). Constraining variables explained 26.26% of variation in CCA. CCA1 and CCA2 were both significant ($p < 0.001$), explaining 31.28% and 28.56% of constrained variation, respectively. DO and pH were significant constraints on microbial community beta diversity ($p < 0.001$), as was temperature to a lesser degree ($p < 0.05$). DOC, however, was not found to be significantly related to microbial community beta diversity. Similarly, *Jones, Newton & McMahon (2009)* found

that DOC concentration does not predict microbial community structural differences, but rather quality of organic carbon (as measured by water color: chlorophyll-a) is significantly related to microbial community structure in freshwater lakes. The influence of DO on microbial community structure is of particular interest due to oxygen's influence on regulation of redox cycles within aquatic systems. As such, partial CCA (pCCA) examining the strength of dissolved oxygen as an environmental constraint on microbial community structure was accomplished while controlling for temperature and pH within sampled lakes. Partial CCA found that oxygen alone was significantly related to microbial community composition ($p < 0.001$, Fig. S2) irrespective of the influence of pH and temperature. The lakes sampled within our study were all located on (or near) Beaver Island within 17 km of each other, yet they were physicochemically diverse, suggesting that environmental constraints on microbial communities are stronger than geographic distance. These results are consistent with established theory that microbial community structure and taxa can be highly constrained by environmental factors within lakes, while geographic proximity of lakes may explain to a lesser degree microbial community structure (*Yannarell & Triplett, 2005*; *Van der Gucht et al., 2007*).

Surface and bottom-water microbial communities within lakes that experienced oxygen and pH stratification (Lake Geneserath and Fox Lake) separated over time (Fig. 4). These results are consistent with previous research that has found divergence of microbial community beta diversity between epilimnia and hypolimnia after lake mixing events (*Shade, Chiu & McMahon, 2010a*), particularly in relation to differences in oxygen as a strong constraint (*Shade, Jones & McMahon, 2008*; *Shade et al., 2011*). Interestingly, while surface-water microbial communities remained relatively stable within these stratifying lakes, bottom-water communities showed marked divergence over time. Previous studies have illustrated that hypolimnetic communities are not resistant to disturbances, particularly disturbances related to oxygen or key nutrient shifts (*Allison & Martiny, 2008*; *Shade et al., 2011*; *Shade et al., 2012b*). Our results corroborate that lake stratification may be an important factor in shaping these communities across freshwater lakes which experience water column mixing events. Contrastingly, within lakes which did not experience stratification, community composition was indistinguishable between surface-water and bottom-water communities within each lake respectively. Previous research has found that microbial communities within oxygenated hypolimnia of Lake Michigan and other deep lakes are often structurally distinct from the respective epilimnia (*Fujimoto et al., 2016*; *Okazaki et al., 2017*). It is likely that we did not find distinctness between surface water and bottom water communities of Lake Michigan, as the hypolimnion of Lake Michigan was not sampled within this study. The Lake Michigan sampling point was also shallower in depth than locations explored by *Okazaki et al. (2017)* and a separate location than studied by *Fujimoto et al. (2016)*.

## Taxonomic relationships to environmental variables

Nine hundred twenty-eight microbial OTUs were found within a minimum of five samples and these shared OTUs were analyzed for correlations with measured environmental variables. In general, specific taxonomic groups (at the level of genus or higher) appeared

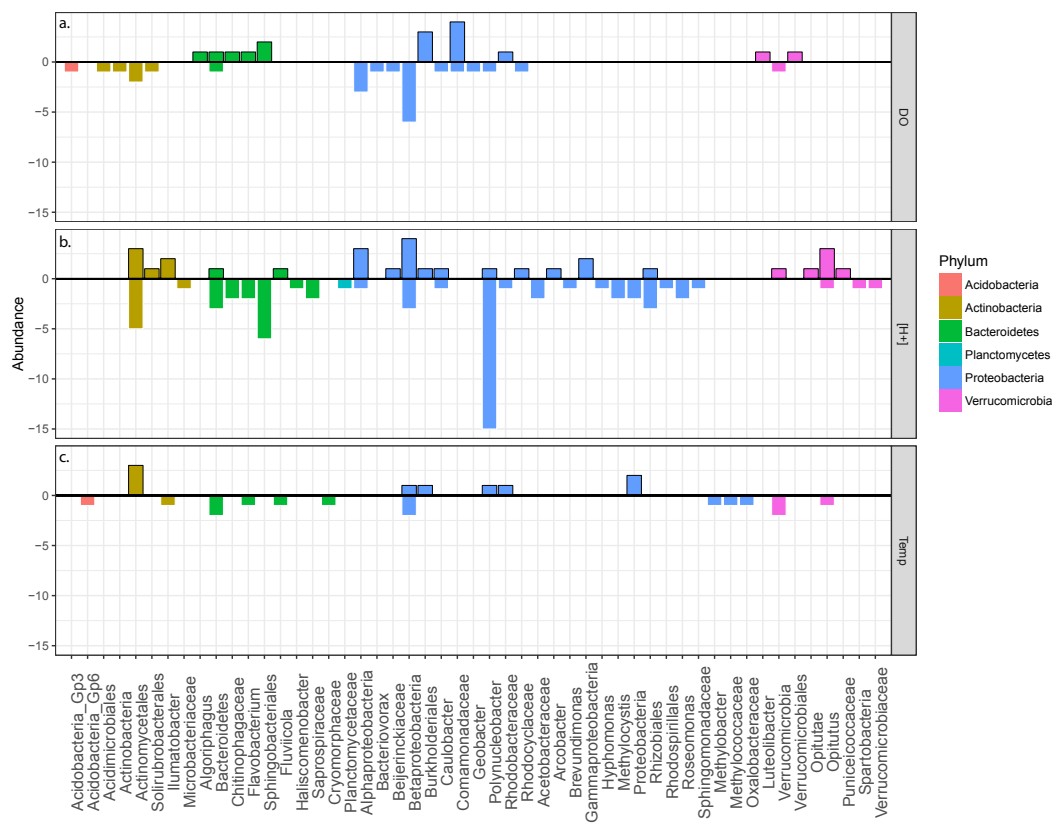

**Figure 5  Bar plot of taxonomic groups.** Bar plot illustrating correlations of taxonomic groups (identified to their lowest phylogenetic classification) to environmental variables (A) dissolved oxygen (DO), (B) pH ([H +]), and (C) temperature (Temp). Abundance values indicate number of OTUs identified to a taxonomic group either positively or negatively correlated to an environmental variable. Values above the "0" line indicate positively correlated OTU abundances, while values below this same line represent negatively correlated OTU abundances. Bar colors correspond to the phylum each lower taxonomic group belongs to.

to be either positively or negatively correlated to levels of dissolved oxygen, as there was little contrast in correlation direction from the same taxonomic group (Fig. 5A). Specifically, members of the phylum *Bacteroidetes* were primarily positively correlated with dissolved oxygen ($n = 6$), where only one *Bacteroidetes* OTU was negatively related to dissolved oxygen. Within *Bacteroidetes*, a representative OTU from the genus *Algoriphagus*, which has been described as a strict aerobe (*Bowman, Nichols & Gibson, 2003*; *Liu et al., 2009*), was found to positively correlate with dissolved oxygen concentrations. Members of *Flavobacteria* have been known to be primarily aerobic (*Bernardet et al., 1996*), and were also found to positively correlate to dissolved oxygen concentration. Other OTUs, related to *Sphingobacteriales* (including *Chitinophagaceae*), contain representative aerobic microbial taxa (*Rosenberg, 2014*) and are common in freshwater bodies within the Great Lakes basin (*Mou et al., 2013*), so it is not surprising to find these taxa within aerobic freshwater environments within the temperate freshwater lakes of the Great Lakes region. *Comamonadaceae* were generally positively related with dissolved oxygen levels, however,

one representative was negatively correlated. Research has demonstrated that some members of this primarily aerobic family indeed can grow under anaerobic conditions (*Ramana & Sasikala, 2009*). Many OTUs related to the order *Burkholderiales* also increased with oxygen availability. It is likely that these taxa are unable to adapt to developing anoxia within the hypolimnion of chemically stratifying lakes, and may play more dominant roles within the epilimnion after stratification has occurred post-mixing.

To the contrary, several individual OTUs negatively correlated with dissolved oxygen. Members of the phyla *Acidobacteria*, *Actinobacteria*, *Alphaproteobacteria*, and *Deltaproteobacteria* only had representative OTUs found to be inversely correlated to dissolved oxygen concentrations. Representative OTUs from *Alphaproteobacteria* included taxa related to *Caulobacter* and *Rhodocyclaceae*, both of which are bacteria that could thrive under anaerobic conditions (*Song et al., 2013*; *Oren, 2014*). From *Deltaproteobacteria*, *Geobacter*, a well-renowned anaerobe (*Lovley & Phillips, 1988*; *Lovley et al., 1999*), was found to be negatively correlated to oxygen, as was *Bacteriovorax*, of which much less is known regarding its metabolism in freshwater systems. These results point towards taxa that may prosper in developing anaerobic hypolimnetic environments after a lake mixing event has disturbed the water column.

Frequently, OTUs were idiosyncratic in their relationships to higher or lower pH levels within the same Phylum and ranging down to Genus (Fig. 5B). This suggests that preferences for ideal environmental pH are often at the level of OTU, and generalizations cannot be drawn for many taxonomic groups. Despite this, there were groups of bacteria that correlated predominantly with decreasing $[H^+]$, with few or no representative OTUs correlating with increasing $[H^+]$. For example, *Polynucleobacter* OTUs almost resoundingly correlated to decreasing $[H^+]$, despite previous research suggesting that members within this genus comprise a higher proportion of microbial communities within environments characterized by circumneutral to acidic pH (*Jezbera et al., 2012*). It is possible that these taxa may have been constrained by other factors (such as DOM or lack of $O_2$), which limited them from thriving within lower pH environments often corresponding with lower $O_2$ levels. From the phylum *Bacteroidetes*, OTUs related to *Chitinophagaceae*, *Flavobacterium*, and *Sphingobacteriales* negatively correlated to $[H^+]$, as did *Proteobacteria* members such as *Acetobacteraceae*, *Hyphomonas*, *Methylobacter*, and *Roseomonas*. As pH generally decreases with increasing depth within a water column, it could be superficially suggested that these taxa may be more abundant in shallower depths of the water column. As an example, *Bacteroidetes* which negatively correlated to $[H^+]$ contained OTUs which positively correlated to dissolved oxygen, suggesting that these OTUs are likely present within epilimnia of stratified lakes. Interestingly, the family *Acetobacteraceae*, which contains members of the acetic acid bacteria (including *Roseomonas*), are often adapted to lower pH levels due to their ability to produce acetic acid during metabolism (*Raspor & Goranovič, 2008*). However, members within this group are obligate aerobes (*Raspor & Goranovič, 2008*), and thus may have been unable to tolerate lower $O_2$ conditions as may have been the case for OTUs related to *Polynucleobacter*.

Temperature did not appear to have a large influence on individual microbial taxa within these lakes relative to the potential influences of pH and dissolved oxygen (Fig. 5C).

However, individual OTUs spread across several phyla periodically correlated with temperature either positively or negatively. Most notably, *Actinomycetales*, which possess thermophilic taxa (*Korn-Wendisch et al., 1995*), contained three OTUs which positively correlated to temperature, suggesting that these taxa may be most prevalent in shallow lakes or epilimnia.

## CONCLUSION

This study has found that microbial communities within actively physicochemically stratifying lakes, particularly stratification of dissolved oxygen, pH, and temperature, diverge to a larger degree over time relative to communities within lakes (or points within lakes) that do not chemically stratify. Additionally, despite their relatively close geographic proximity, each lake harbored a distinct microbial community, suggesting that lake physicochemistry is a stronger constraint on microbial communities than geographic region. Correlations of individual microbial OTUs to physical and chemical variables, such as dissolved oxygen, pH, and temperature, could be related to metabolic capabilities of microbial taxonomic groups or individual OTUs. This suggests that lake stratification and environmental conditions unique to each lake may influence the prevalence of some microbial taxa more strongly than others, thereby potentially influencing ecosystem processes carried out by these taxa. This research highlights the importance of sampling lakes in the same geographic area but distinct in physical and chemical attributes, as well as the potential impact of lake mixing and stratification as a disturbance to microbial communities within temperate freshwater lake systems, which could ultimately influence microbial community functional diversity and biogeochemical processes.

## ACKNOWLEDGEMENTS

Special thanks to Stephen Lokos, Alana Miles, Greg Kinney, Dave Schuberg, and John Gordon for their assistance during sampling weeks. This paper is Contribution Number 89 of the Central Michigan University Institute for Great Lakes Research.

### Funding

This work was supported by the CMU Great Lakes Summer Undergraduate Research program and the College of Science and Engineering. The funders had no role in study design, data collection and analysis, decision to publish, or preparation of the manuscript.

### Grant Disclosures

The following grant information was disclosed by the authors:
CMU Great Lakes Summer Undergraduate Research program.
College of Science and Engineering.

### Competing Interests

The authors declare there are no competing interests.

## Author Contributions

- Miranda H. Hengy and Dean J. Horton performed the experiments, analyzed the data, wrote the paper, prepared figures and/or tables, reviewed drafts of the paper.
- Donald G. Uzarski conceived and designed the experiments, analyzed the data, wrote the paper, reviewed drafts of the paper.
- Deric R. Learman conceived and designed the experiments, analyzed the data, contributed reagents/materials/analysis tools, wrote the paper, reviewed drafts of the paper.

## DNA Deposition

The following information was supplied regarding the deposition of DNA sequences:

Sequences obtained for this study have been deposited in the MG-RAST database (*Meyer et al., 2008*) under accession numbers mgm4732740.3–mgm4732751.3, mgm4732757.3, mgm4732760.3, mgm4733677.3–mgm4733686.3, mgm4733688.3, mgm4733690.3–mgm4733704.3, and mgm4733784.3–mgm4733785.3.

## Data Availability

Github: https://github.com/horto2dj/CMUBS_microb.

## Supplemental Information

Supplemental information for this article can be found online at http://dx.doi.org/10.7717/peerj.3937#supplemental-information.

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
