# Peer review of "Microbial community diversity patterns are related to physical and chemical differences among temperate lakes near Beaver Island, MI"

_PeerJ, doi:10.7717/peerj.3937_

## Round 0.1 · original submission · Major Revisions

· Academic Editor

Major Revisions

2 reviewers agree that the work has merit but needs major revisions, while a third went for Minor Revisions. All reviewers agree that the problems are in the experimental design and how to address something as ambitious as the title announces.

However, in his paper Hengy et al. describe the microbial diversity of four lakes with different physical and chemical characteristics. They describe the environmental differences between and within lakes (different depths) at three different times and correlate them with the microbial community structure. Even though the sample sizes are not large I think the results are clear and concise. This study contributes to the knowledge on the potential effect that environmental variables have on the bacterial community dynamics in aquatic habitats.

·

Basic reporting

In this paper Hengy et al. describe the microbial diversity of four lakes with different physical and chemical characteristics. They describe the environmental differences between and within lakes (different depths) at three different times and correlate them with the microbial community structure. Even though the sample sizes are not large I think the results are clear and concise. This study contributes to the knowledge on the potential effect that environmental variables have on the bacterial community dynamics in aquatic habitats.

Experimental design

Overall, the experimental design is correct. However, in the general comments section you will find some suggestions and aspect that need to be clarified.

Validity of the findings

In the general comments section you will find my main concerns which have to do with rarefying the sequence data and correcting for multiple comparisons when calculating correlations.

Additional comments

Below you will find some general or specific suggestions that are mainly aspects associated to the methods.

Line 73: Change “reaction” to “reactions”
Lines 102-103: I assume that the water sample was passed through the larger filter first and then the smaller one? Please explain with more detail.
Line 118: Replace “was measured using and an ion…” by “was measured using an ion…”
Lines 130-131: This sentence is not clear. Was the DNA extracted from both filters at the same time? What is the reason to pool both filters? The 2.2 filter excludes all large particles and cells so I would think the 0.22 is the one that should have all the bacterial cells. Please explain.
L134: the 27F and 1493R primers amplify the whole 16S rRNA gene, not only the V4 region. Please correct this sentence.
Lines 142-151: You need to give more details on the processing of the sequences on this section and some also on the results section:
Include how many sequences were obtained initially.
Which filtering steps did you do in addition to the analysis of chimeras and how many sequences were analyzed after the filtering steps
How many OTUs were obtained?
Did you rarefied the samples? At which depth? If the authors didn’t rarefied the samples I strongly suggest to do it.
Line 154: Change “Statistical analysis (both chemical and biological) was completed” to “Statistical analyses (both chemical and biological) were completed”
Line 153: For this section please clarify why did you retain singletons for alpha diversity and removed them for beta diversity. The same OTU table should be used for both diversity estimates. Otherwise the results may be biased. Also, this section could be organized better.
Lines 155-158: This sentence should be included in the “Microbial Taxonomic Analysis” section of the methods.
Lines 158-160: Move this sentence to line 155
Lines 169-171: How were the samples normalized? Explain the variance transformation.
Lines 185-191. Did the authors correct for multiple comparisons? If not I strongly suggest to do it. Otherwise you might be overestimating the significant correlations.
Line 195: Change “was observable” to “were observed”
Lines 211-212: Move this sentences to methods. Also it is not clear what the authors mean with relative abundance. Do you mean sequencing depth?
Lines 210-224: Is it very important to clarify if the authors rarefied the samples (OTUs) in order to compare diversity values across samples. Otherwise the alpha diversity estimates may be biased.
Lines 281-282: This information should be included at the beginning of the results section. You could include how many OTUs were shared among lakes and how many OTUs were unique to each lake or unique to groups of lakes that are more similar in terms of the environmental variables.

I suggest including a map of the lakes sampled as an additional figure.
Figure 1: You could add a legend for the habitat as is done in the other figures
Figure 2: could be improved, so that the legend can be distinguished clearly. The labels on the X axis could be larger and the bars could be thicker. Also I would suggest to change the colors so that it’s easier to distinguish the different bars

Reviewer 2 ·

Basic reporting

Writing should be improved for clarity.
- The emphasis on 'lakes within the Laurentian Great Lakes basin' in the title and abstract is misleading. The Great Lakes basin is massive – 4 lakes is too few lakes to characterize lakes in the Great Lakes basin and it sounds like the study lakes are close together geographically near Beaver Island, MI, making them not especially representative of the extent of the Great Lakes basin. I suggest specifying Beaver Island, MI, not the Great Lakes basin, as the location of the study lakes and being more specific in the abstract concerning the number of lakes sampled.
- You claim to show that conditions in the water column ‘significantly impact’ community structure, yet all of your analyses are correlation based. It is more accurate and clear to emphasize the correlations observed in this study than to imply causation.
- Additional examples where clarity could be improved:
--‘physical, chemical, biological, and temporal processes’ in the first line of the abstract does not make sense to me. Physical, chemical, and biological processes may be temporally variable or dynamic, but I do not think that that there are ‘temporal processes’ that are separate from physical, chemical, and biological processes.
--In the next sentence you talk about ‘similar geographic regions’ – do you mean the same geographic region?
--It would help to be more explicit about the differences between DOC and DOM (and to define them before using the acronyms) and what information you gain from looking at both instead of one or the other (lines 50-51 of the introduction)

Suggestions for improvement on literature references and background/ context:
- You need to include a citation for the R statistical environment (type “citation()’” into R to generate the citation – example from the version of R I use on my computer:
R Core Team (2016). R: A language and environment for statistical computing. R Foundation for Statistical Computing, Vienna, Austria. URL https://www.R-project.org/.
- The conclusion that ‘despite geographic proximity, each lake harbored a distinct microbial community, suggesting that lake chemistry is a stronger constraint on microbial communities than geographic region’ covers a topic that has previously been explored in freshwater lakes, but it seems like there were a number of relevant studies overlooked from the discussion. I suggest that the authors consider whether previous surveys of lakes provide relevant context to their work, including:
Yannarell and Triplett 2005. Geographic and Environmental Sources of Variation in Lake Bacterial Community Composition. http://aem.asm.org/content/71/1/227
Van der Gucht et al. 2007. The power of species sorting: Local factors drive bacterial community composition over a wide range of spatial scales. http://www.pnas.org/content/104/51/20404.short
- I additionally recommend that the authors consider whether Newton and colleagues (2011) A guide to the natural history of freshwater lake bacteria may provide additional useful context for their sequence data: https://www.ncbi.nlm.nih.gov/pubmed/21372319. In my opinion, the freshwater database initially curated as part of the Newton review is the best database to use when classifying freshwater sequences (though whether sequence database really matters depends on your question). The database and instructions on how to use it can be found here:
https://github.com/mcmahon-uw/FWMFG

I commend the authors for including accession numbers for their sequence data. However, I am not sure whether there is sufficient metadata available with your deposited sequences. From what I was able to see when I looked up your accession numbers in MG-RAST, I can tell the location but not the depth or time sampled. A key may be needed to connect name to MG-RAST ID to necessary metadata, perhaps this could be included as a supplemental table.

While not necessary for publication, I think it would be useful if the R code, OTU table, and metadata were shared via GitHub.

The article structure is professional, though I do have a few suggestions that might improve the presentation of the study sites and results. I think it would be useful to include a table in the main text with the basic limnological characteristics of each lake (e.g., latitude and longitude, ranges of observed water temperature and chemistry parameters described in supplemental table 2). I am not sure what the value is in including an NMDS plot as well as CCA/ pCCA plots. Since constrained ordinations force samples to correspond to measured environmental parameters, I am not sure how useful it is to include these plots in the main text - perhaps they would be more appropriate as supplemental figures.

Experimental design

The authors present original primary research that appears to be within the aims and scope of PeerJ.

I am unclear on what knowledge gap is specifically being filled. I think the research question could be more clearly defined.

The epilimnon/ hypolimnion contrast is interesting; however, it does not appear that the hypolimnion of Lake Michigan was sampled. It is not unusual for the mixed layer depth in Lake Michigan to exceed the 14.5-18.3 m from which bottom samples were collected. Temperature profile data from supplemental table 1 supports the idea that the Lake Michigan hypolimnion was not sampled.

The sequencing methods are not described in sufficient detail. First, 27F to 1492R primers span almost the entire 16S rRNA gene, encompassing far more than the V4 region. Does this mean that nested PCR was used? What primers were used for sequencing? If the 27F and 1492R primers were the primers used for sequencing, the MiSeq reads could not have merged. If this was the case, did you only analyze forward or reverse reads?

If the number of cycles were varied for each cycle, you should include somewhere, perhaps in a supplemental table how many cycles were used for each sample.

How long were your Illumina MiSeq reads (e.g., 2 x 250)?

You cite Kozich et al. and say that sequences were processed using Mothur, but if you followed the Mothur MiSeq protocol you should specifically state that you followed the Mothur MiSeq SOP and include the date that you accessed the SOP. Any deviations made from the Mothur MiSeq SOP (if this is what you followed) should be specifically noted.

What was the rationale behind removing singletons prior to normalizing samples for beta diversity analyses? I am not aware of this being a standard practice, and it does not seem like a good idea to me.

Validity of the findings

Some of the conclusions may be overstated. For example, stating that the data 'show that differences in phyicochemical conditions in the water column can significantly impact aquatic microbial community structure' seems to imply causation when the study was correlational and involved few lakes.

It is challenging for me to follow the links between the original research question, supporting results, and conclusions.

I find some statements in the results/ discussion to be confusing. For example, the statement starting on line 269 that community composition was indistinguishable between surface water and bottom water communities when oxygen is not stratified seems like a sampling artifact to me. Lake Michigan surface communities from open water stations in the middle of the lake differ significantly from communities sampled from the hypolimnion even though the hypolimnion is oxygenated. A recent paper published in ISME J has also described differences between surface and hypolimnion communities in lakes where the hypolimnion is oxygenated: http://www.nature.com/ismej/journal/vaop/ncurrent/full/ismej201789a.html?WT.feed_name=subjects_microbiology.

Additional comments

What does it mean to explore microbial communities within lakes on an 'individual level' - one lake at a time (line 38)?

It is not clear to me how relationships between taxa and environmental gradients are being 'deeply' explored (line 84). Is this a reference to sequencing depth?

I am not sure that I agree with the interpretation of Jezbera et al. 2012 lines 317-319; Polynucleobacter comprised 1.1-5% of the bacterial assemblage in most lakes with pH between 8.1 and 8.5.

I am not sure what it means when Polynucleobacter is described as a 'generalist' genus (line 233). It makes it sound like members within the genus are generalists, when I think the reality is that the members of Polynucleobacter are quite diverse.

Reviewer 3 ·

Basic reporting

This paper is clearly written and relevant, but there are numerous weaknesses that I think preclude it from being published in its current state.

With respect to literature cites, the authors should consider referencing other comprehensive 16S characterizations of freshwater lakes, particularly ones in the Great Lakes (e.g., Fujimoto, Masanori, et al. "Spatiotemporal distribution of bacterioplankton functional groups along a freshwater estuary to pelagic gradient in Lake Michigan." Journal of Great Lakes Research 42.5 (2016): 1036-1048)

I appreciate the wide array of statistical tests applied, but I don't understand why they are all relevant. Specifically, the NMDS and CCA plots seem redundant and it doesn't seem meaningful for this particular paper to do both, when they are specifically interested in constraining their datasets to the variables they measured. The application of different tests does reads like an after thought, not as something planned with their study. Thus, the data are seemingly made to support their hypothesis, however the experimental design is weak (see below).

With respect to the taxa and comparisons to other studies, the authors should use the freshwater 16S database (https://github.com/mcmahon-uw/FWMFG).

Experimental design

There are no sample replicates, which is, in my mind, a huge weakness. The authors are making vast generalizations about an entire lake based on 120 ml. They also reference "seasonal" differences (line 95 and elsewhere), but all samples were taken in the summer within a few weeks of each other.

The PCR conditions are also problematic; 36-40 cycles is a lot. The Earth Microbiome Project recommends 35 (or fewer) and many papers, past and present, show issues with high cycle numbers (e.g., Qi et al., AEM 2001), especially depending on which Taq was used (which the authors do not report here).

Validity of the findings

I only question the findings because of the experimental limitations (lack of replicates, post hoc decisions of statistical tests, and PCR/database issues. I really think they should tone down their findings should this get published. That said, I think the findings fit the data and tests that they have.

Additional comments

Line 138: "PCR reactions" is redundant
Line 197: So4 is not shown on Fig 1
Fig. 1: "temp" is cut off and abbreviations should be indicated in the figure legend
Lines 202-3: Lake Michigan is also stratified in the summer
Paragraph beginning on line 257: There are numerous contradictory statements here. I think the authors mean to discuss the lack of an oxycline in Lake Michigan, but instead they say that there was no divergence between LM deep/surface communities, which is stratified, but lake stratification is important in structuring communities.
Line 211-12: I don't understand what these numbers mean in terms of relative abundance.
Lines 219-222: These two sentences are contradictory.
Line 274+ What unmeasured environmental differences? This is very vague.

---

## Round 0.2 · Minor Revisions

· Academic Editor

Minor Revisions

I believe the authors have addressed many of the the reviewers concerns and have done this carefully. The paper reads better and is a good contribution on freshwater microbial ecology. As you can see from the 3 re-reviews, however, there are still many issues which should be addressed in a revision.

·

Basic reporting

The new manuscript has greatly improved in clarity and has gained strength by adding additional references and clarifying several aspect of the methods and results.

Experimental design

I agree with the experimental design but I have made specific suggestions regarding alpha and beta diversity analyses (see general comments below).

Validity of the findings

Data and results are clear. Conclusions are well written.

Additional comments

I agree with all the modifications that the authors made to the manuscript based on the reviewers’ comments. I think that manuscript has greatly improved.

However, I still have one concern regarding alpha and beta diversity analyses. My self (and other reviewers) had previously asked about why the authors retained singletons for alpha diversity and removed them for beta diversity.

The authors gave the following response:
“Singletons were retained for alpha diversity as the chao1 index relies on singleton abundances for proper implementation of the chao1. However, as singletons are often removed to calculate beta diversity estimates and visualization, as many singletons could be sequencing artefacts.”

Based on this response I still think this is not the best approach for two reasons: (1) Chao1 index is presented on table 2 but it’s not even mentioned in the results and discussion and the authors rely only on the Shannon index results. Therefore, I do not see the need of keeping chao1 in the manuscript and therefore I see no need in keeping singletons. (2) the authors stated that singletons could be artifacts so I do not see the relevance of keeping them at all.

The authors should eliminate the singletons and (as I said in my first review) they should use the same OTU table for both diversity estimates.

Reviewer 2 ·

Basic reporting

Basic reporting has been improved. I commend the authors for including additional metadata and access to their analysis scripts through github. There are a few lingering issues needing to be resolved before the manuscript is suitable for publication.

There is still remnant language implying causation, for example:
'driven by' in the title
line 30 - 'can significantly impact'

There are also descriptions obscuring the fact that the hypolimnion of Lake Michigan was not sampled, for example:
line 31-32 - hypolimnion of Lake Michigan was not sampled; most accurate representation would be 'surface' and 'deep'

line 371 - how deep did Fujimoto et al sample? It is likely that you observed differences because you were not sampling analogous deep samples; I do not think your results are directly comparable due to depth differences and the fact that you did not sample the hypolimnion.

line 473-476 - you did not sample the hypolimnion of Lake Michigan and need to be clear about this point

Description of study in the final introductory paragraph needs to be edited for clarity (lines 153-156)
1) 4 lakes, 3 holomictic... does this mean #4 is meromictic? Why would you not say something about lake #4?
2) Lake Michigan did not experience "temperature stratification, but did not stratify"; Lake Michigan was stratified; you did not sample a site that exhibited stratification

line 430 - need more explanation as to why your results differed from Jones et al 2009. I'm guessing they surveyed lakes covering a much broader range of DOC than you did.

Table 1 - There should be separate entries for surface and deep samples; this would improve clarity.

Figure 5 - CCA is most useful for the statistics it generates; plots are typically uninformative. I recommend moving CCA plots to supplemental.

Minor comments related to language and clarity:
Title suggestion: 'near' instead of 'within' Beaver Island, MI

line 37 - temperature is a physical parameter, not chemical

line 39 - constrained analyses demonstrated that DOC did not constrain? (what does that mean?)

lines 115-119 - run-on sentence; break up into more than one sentence

line 410 - 'stratified' instead of 'stratifying'?

line 412 - what about 'Relationship between environmental conditions and microbial beta diversity'? Current phrasing is not clear to me.

line 578 - I don't think 'mitigate' fits here... perhaps 'carried out'?

Experimental design

Necessary details are still missing:
line 250 - explicitly state the primers used for sequencing
line 254 - 'implemented similarly' should be replaced by something to the effect of 'following the Mothur MiSeq SOP with the following modifications' (and then in your brief description, be sure to describe every deviation from the SOP); you need to include the date you accessed the SOP as per the instructions on the website (SOP is not static and has changed over the last few years)
line 257 - what version of Silva?
line 265 - what version of RDP?

Multiple reviewers questioned the removal of singletons and doubletons prior to calculating beta diversity (line 301); rationale and citation provided to justify approach in response to reviewers were not sufficient in my opinion. I recommend re-doing this analysis without removing singletons and doubletons or finding an article that articulates a rationale for this approach, not simply one that went ahead and also removed rarely observed sequences.

Validity of the findings

Validity of the findings is tied to some of the issues described above. I think improving clarity of basic reporting will improve the validity of the findings.

Reviewer 3 ·

Basic reporting

The manuscript is improved from the original version. The manuscript is clearly written, uses appropriate citations and background, and the figures and tables are all sufficient.

Line 108: This does not make sense: "...experienced temperature stratification but did not stratify."

Line 124: How deep was Lake Michigan at the site where you sampled? The depth at which you sampled is not even below the thermocline.

Experimental design

I still find some flaw in the design that there is only one sample per lake per depth per time. To extrapolate to whole lake conditions on a couple of liters of water is somewhat problematic, especially since the authors point out that there's lake heterogeneity.

Line 176: Methods are still unclear. Aside from the exceptionally high cycle number used to amplify the full length 16S gene, you then amplified the V4 region specifically from these first amplicons? How many cycles did that add? Do you have references that support using this approach, because it is somewhat different than the standard 16S sequencing protocols?

I also do not understand why the authors are so reluctant to classify their sequences using the freshwater 16S database; this would provide more information and better taxonomy than RDP.

Validity of the findings

Paragraph beginning line 315: the observations/discussion seems to have some logical flaws. The authors claim that these lakes diverge over time and compare that to observations during lake mixing; however, these lakes remained stratified and, in fact, became more (oxygen) stratified over time. In fact, of the variables that were measured, oxygen is the only one to change between the the three timepoints, so it is no surprise that this is what can be linked to the change in community structure. Perhaps the authors should discuss what drives the shift to anoxic bottom waters as well.

---

## Round 0.3 · accepted · Accept

· Academic Editor

Accept

The paper now reads more fluidly and all the concerns of the reviewers have been addressed properly. Thanks for your patience.